# The Role of Complement in the Pathogenesis and Treatment of Myasthenia Gravis

**DOI:** 10.3390/cells14100739

**Published:** 2025-05-19

**Authors:** Armando Martinez Salazar, Sepideh Mokhtari, Edwin Peguero, Muhammad Jaffer

**Affiliations:** 1Department of Neurology, Morsani College of Medicine, University of South Florida, Tampa, FL 33612, USA; amartinezsalazar@usf.edu (A.M.S.); sepideh.mokhtari@moffitt.org (S.M.); edwin.peguero@moffitt.org (E.P.); 2Department of Neuro-Oncology, Moffitt Cancer Center, Tampa, FL 33612, USA

**Keywords:** myasthenia gravis, complement, acetylcholine receptor antibodies, zilucoplan, eculizumab, ravulizumab, gefurulimab, cemdisiran, pozelimab, vemircopan

## Abstract

Myasthenia gravis is an antibody-mediated autoimmune condition characterized by defects in cholinergic transmission at the neuromuscular junction. In AchR antibody-positive patients, complement activation plays a prominent role in the disease process, which appears to be mediated by the activation of the membrane attack complex. Since IgG4 is not a good complement activator, the role of complement in MuSK antibody-positive myasthenia gravis patients is negligible. Experimental animal models of myasthenia gravis have shown promise with the antagonism of different elements of the complement cascade, with positive clinical outcomes. This has led to the development of the first C5 inhibitors approved for myasthenia gravis with AchR antibodies: eculizumab, ravulizumab, and zilucoplan. Other clinical trials are currently in progress, investigating the potential therapeutic role of other targets, including the Factor B inhibition or hepatic synthesis of the C5 protein. Other proposed potential targets that have not yet been clinically tested are also discussed in this review article.

## 1. Introduction

Myasthenia gravis (MG) is a classic example of an antibody-mediated neuroimmune disorder of defective cholinergic transmission at the neuromuscular junction of skeletal muscles. The incidence of the disease is approximately 30 cases per million person-years, while its prevalence is approximately 200 cases per million [1]. The disease is mediated by IgG subtype autoantibodies that result in a class II hypersensitivity reaction at the post-synaptic junction, directly causing end-organ damage. This often manifests in variable degrees of fatigable weakness in ocular, neck, limb, bulbar, and respiratory musculature; the disease severity can vary from limited ocular muscle involvement to life-threatening respiratory failure. Mortality has improved with the advent of endotracheal intubation; however, this still persists at around 5% [2].

The most implicated autoantibody is against the nicotinic acetylcholine receptor (AchR) in approximately 80% of cases, followed by muscle-specific kinase antibodies (MuSK) in 5–10% of cases. Other autoantibody targets exist in about 10% of cases and include low-density lipoprotein receptor-related protein 4 (Lrp4), agrin, titin, ryanodine receptors, rapsyn, cortactin, collagen Q, Kv1.4, and collagen XII [3].

AchR antibodies bind to the potently immunogenic AchR, a transmembrane glycoprotein in the junctional folds of post-synaptic muscle responsible for acetylcholine-dependent depolarization. These antibodies are predominantly bivalent of the IgG1 and IgG3 subclasses and impair receptor function by either the blockade of AChR channel function, the cross-linking of AChR by the divalent AChR antibody leading to increased endocytosis and degradation (antigenic modulation), or complement activation. This third mechanism is the subject of this review article. Complement activation in turn precipitates the formation of the membrane attack complex, leading to damage of the junctional folds, degradation of the AchR, and damage of proteins (such as utrophin and rapsyn) that maintain the neuromuscular junction. These changes lead to the restructuring of the neuromuscular junction with the widening of the junctional folds, the simplification of the post-synaptic membrane, and the accumulation of debris in the synaptic clefts [4].

Meanwhile, MuSK autoantibodies are monovalent and predominantly of the IgG4 subclass, lacking the ability to activate the complement cascade; the target tyrosine kinase receptor is responsible for clustering of AChR at the neuromuscular junction. MuSK antibodies bind to the Ig-like domain of the MuSK protein, preventing its phosphorylation and subsequently disrupting the Agrin-Lrp4-MuSK-Dok-7 signaling pathway. The targeting of Dok-7, a muscle-intrinsic activator of MuSK required for synaptogenesis, reduces the density of AchR in the post-synaptic membrane [5]. Similarly, low-density lipoprotein receptor-related protein 4 (LRP4) antibodies are primarily IgG1 and IgG2 subtypes, inhibiting the clustering of AChR, and these also appear not to strongly activate complement, perhaps because of a relatively higher concentration of IgG2 in LRP4 antibodies [6]. LRP4 is the post-synaptic receptor of agrin, and the blocking of this protein directly disrupts the activation of MuSK, in turn causing a reduction in AchR clustering [7]. Patients with other autoantibodies are highly heterogeneous, and their exact pathophysiological mechanisms remain under active investigation.

## 2. Complement Pathways and Components

The complement cascade is part of the innate immune system and consists of approximately thirty proteins that operate via a series of proteolytic cleavages that lead to the assembly of convertases into the membrane attack complex (MAC). This system is responsible for acute inflammation, microbial opsonization, and the solubilization of antigen–antibody complexes. The cascade is activated either by antibodies (classical pathway), spontaneously formed C3b (alternative pathway) either due to foreign pathogens or polymeric IgA, or the binding of proteins on bacterial cell surfaces (mannan-binding lectin pathway). These pathways depend on different molecules for their initiation, but they converge to generate the same set of effector molecules [8].

The classical pathway is activated when IgG1 or IgG3 autoantibodies converge at their Fc domains to C1q; this subsequently leads to the autoactivation of C1r, leading to the activation of the C1 complex. C1 then cleaves C4 into two subcomponents, C4a and C4b. C1s and C1r then fuse with C4b to form C14b. The amplification phase occurs when C14b enzymatically cleaves C2 to C2a and C2b. C14b combines with C2a to form C14b2a, which is also known as C3 convertase.

Meanwhile, in the alternative pathway, the spontaneous hydrolysis of C3 may also occur, and the formation of C3b combined with Factor B produces C3 convertase. C3 convertase enzymatically cleaves C3 into C3a and C3b. C3b with C3 convertase forms C14b2a3b, which is C5 convertase. C5 convertase then cleaves C5 into C5a and C5b. C5b then fuses with C6, C7, C8, and C9 to create C5b6789, which is the MAC effector mechanism of the complement system.

Mannan-binding lectin, like C1q, is a six-headed molecule that forms a complex with two protease zymogens, which, in the case of the mannan-binding lectin complex (MBL complex), are MASP-1 and MASP-2. When the MBL complex binds to a pathogen surface, MASP-1 and MASP-2 are activated to cleave C4 and C2. Thus, the MB-lectin pathway initiates complement activation in the same way as the classical pathway, forming a C3 convertase from C2b bound to C4b.

These various complement pathways and their effector functions are summarized below in Figure 1. There is a significant body of evidence to support the role of the classical pathway in complement activation in the pathogenesis of myasthenia gravis, with the other two pathways implicated less frequently [9]. For instance, histopathological studies have demonstrated the presence of C3 and C9 within the neuromuscular junction of MG patients, substrates for the formation of MAC [10]. Furthermore, murine models have demonstrated that a deficiency of C3, C4, C5, or C6 results in a significantly lower incidence of developing experimental MG [11]. Finally, various studies targeting the complement pathway in murine models with anti-C1q, anti-C5, and anti-C6 antibodies resulted in the inhibition of passively induced experimental autoimmune MG [12].

## 3. Complement-Related Diseases

While the role of complement in the pathogenicity of MG cannot be overstated, it also behooves us to review the role of complement in other autoimmune and immunodeficiency disorders. With disequilibrium in complement proteins, several complement-related diseases may occur. These disorders may be inherited or acquired in nature.

Inherited disorders of complement components present predominantly with recurrent bacterial infections and/or systemic lupus erythematosus (SLE). In particular, deficiencies of C1qrs and C4 result in an SLE-like illness, which is believed to ensue due to autoimmunity resulting from the inadequate clearance of apoptotic cells and immune complexes [13]. Meanwhile, inherited deficiencies in C2, C3, and C5–9 are more highly associated with immunodeficiency disorders and an increased risk of recurrent infections with pneumococcal and neisserial pathogens [14]. For instance, infants with a C5 complement deficiency may develop Leiner disease, or erythroderma desquamativum, which manifests as recurrent diarrhea, wasting, and generalized seborrheic dermatitis [15].

On the other hand, acquired deficiencies in complement proteins are more common than inherited complement disorders. The mechanisms responsible for acquired complement deficiencies include, in order of commonality, the following: accelerated consumption by immune complexes, reduced hepatic synthesis, and loss of complement components in the urine. Hypocomplementemia secondary to accelerated consumption has been implicated in causing SLE, antiphospholipid syndrome, cryoglobulinemia, and various systemic vasculitides [16]. Meanwhile, diseases such as non-alcoholic steatohepatitis, viral hepatitis, and hepatocellular carcinoma result in decreased complement production, thereby predisposing to infection by encapsulated bacteria [17]. Finally, a rare cause of secondary hypocomplementemia is the loss of complement proteins in the urine. Factor D can be reduced in severe nephrotic syndrome, although this laboratory finding is of uncertain clinical significance [18]. Finally, a rare acquired complement disorder can be seen in paroxysmal nocturnal hemoglobinuria, wherein reduced glycosylphosphatidylinositol-linked complement inhibitors CD55 and CD59 result in complement-mediated intravascular hemolysis [19].

## 4. The Role of Complement in Myasthenia Gravis

### 4.1. Preclinical Data

Complement exhibits a direct pathogenic role in AChR antibody-positive myasthenia gravis. Most of the experimental evidence for complement’s role in the neuromuscular junction comes from experimental autoimmune models of myasthenia gravis (EAMG) [9] or passive transfer models of myasthenia gravis (PTMG), which involve either the active sensitization of subject animals to purified AChR, a potent immunogen, or the passive transfer of anti-AChR antibodies [20]. Active models have the advantage of mimicking the pathogenic process occurring in humans, and generally consist of two phases: the early phase (0–7 days) with IgM production and both phagocytic cell recruitment and complement deposition [21] and the late phase (>28 days) with the increased production of IgG and decreased AchR density in skeletal muscle [22]. Passive models, on the other hand, allow for the demonstration of potential therapeutic efficacy within 2–3 days, after which therapeutics may transition to EAMG models [23].

The earliest experimental model of myasthenia gravis was born in 1973, when AChR from the electric organs of the eel *Electrophorus electricus* was used to immunize rabbits against AChR, reproducing myasthenic symptoms [24]. In models using electric eel AChR, only around 1% of the produced antibodies cross-react with the host’s AChR, and this subset is responsible for the clinical manifestations of the disease [25]. EAMGs have been further studied and refined, based on the ideal characteristics an EAMG should have. EAMGs should be robust, not so mild that therapeutic interventions cannot be appropriately assessed, but also not severe enough that they cause significant animal suffering and represent an unrealistically stringent assessment of response to a therapy [25]. It is also relevant to mention that mice have a larger safety factor amplitude than humans [26]; thus, EAMG might not represent a completely accurate picture of the disease in humans.

Complement activation in myasthenia gravis is predominantly an IgG1 process [4]. Monomeric IgG1 is a poor complement activator; thus, the process of complement activation mediated by antibodies depends on the formation of hexameric IgG1 complexes on the cell surface [27]. Cq1 structurally consists of six heads that bind to a CH2 domain in the Fc portion of the immunoglobulin [28]. While IgG1 and IgG3 can be potent complement activators, the former has low-affinity C1q binding sites while the latter has higher-affinity binding sites that are exposed only in a polymeric state [27]. Carbamylation of IgG, which reduces its capacity for polymerization, was shown in a liposomal lysis assay to completely abolish liposome lysis [29]. Compared to IgG1 and IgG3, IgG4 is a poor complement activator. IgG4 differs structurally from other IgG subclasses, and it has been proposed that variations in its CH2 domain might explain its poor complement-activating properties [30]. To further evince this, small variations in the CH2 domain have been shown to have significant effects on the antibody’s complement binding capabilities [31]. Nonetheless, IgG4 is not entirely devoid of complement activation properties: IgG4 molecules with agalactosylated Fab regions have been shown to activate the lectin pathway [32]. Aberrant galactose residues appear to inhibit or negate the activation of the lectin pathway when compared with IgG molecules containing other carbohydrate residues [33]. This complement activation pathway appears to be negligible in the context of myasthenia gravis, however. Overall, these structural differences between antibodies to AChR and MuSK explain why complement inhibitors do not seem to be effective in MuSK antibody-mediated myasthenia gravis, which is predominantly an IgG4-driven process [34].

### 4.2. Clinical Data

#### 4.2.1. Pathophysiology

The clinical manifestations of myasthenia gravis in AChR antibody-positive models depend on the effect that antibody binding has on transmission through the neuromuscular junction. A basic understanding of normal neuromuscular junction physiology is thus required to explore the role of complement in myasthenic pathophysiology. Normal neuromuscular junction transmission involves the release of acetylcholine in three different storage stores: primary (immediately available), secondary (available after 1–2 s; responsible for the increment seen in repetitive nerve stimulation after 4–5 stimulations), and tertiary (in the axon and cell body) [35]. In normal neuromuscular junctions, the end-plate potential is usually higher than the potential required to generate an action potential, which is termed the “safety factor” [36,37]. It appears that AChR antibodies lead to a reduction in both the muscle end-plate potential as well as the threshold for depolarization, which suggests effects on both the AChR as well as the muscle sodium currents [38].

IgG1 and IgG3 can both directly block and promote the internalization of receptors, and cross-linking of antibodies into hexamers promotes complement deposition [39]. Current evidence shows there is a wide variety of antibodies that can activate complement, and within this heterogenous group of acetylcholine antibodies, antibody binding seems to be crucial for complement activation. There is evidence to suggest that the combination of different antibodies, and not single antibodies, induces the more potent activation of complement [40]. The activation of complement results in both the widening and simplification of the synaptic cleft and accumulation of debris, which has been demonstrated in cytochemical and ultrastructural analysis [41,42]. The simplification of muscle membrane junctional folds likely explains (at least partly) the effect of AChR antibodies in both AChR and sodium currents [37], and this appears to be a complement-mediated process (suggested by the fact that MuSK antibodies, which do not activate complement, have shown steady action potential thresholds while also affecting compensatory ACh release, which is usually upregulated in AChR models) [26,43,44]. The presence of C9 complexes appears to be more prominent in regions with more marked structural abnormalities of the junctional folds, further implying a role for complement in causing ultrastructural muscle changes [41]. Inflammatory cells also appear to play a role in this process; however, in Nakano’s histopathological study of 30 patients with myasthenia gravis, the presence and quantity of inflammatory cells did not correlate with clinical severity at the time of muscle biopsy [42]. This suggests that humoral immunity plays a more significant role in myasthenia gravis pathogenesis than direct cellular immunity. Figure 2 demonstrates the formation of MAC, the mechanism by which complement mediates the breakdown of the neuromuscular junction.

#### 4.2.2. Complement as Biomarkers

There are currently no validated biomarkers for monitoring therapeutic response in myasthenia gravis. Given its role in the pathogenesis of myasthenia, complement has been explored as a potential candidate for prognostication and treatment monitoring [45]. Stascheit et al. evaluated complement levels in cohorts of AChR antibody MG, MusK antibody MG, seronegative myasthenia gravis, and healthy donors. Both complement levels involved in the classical and the alternative pathway were increased (C3a, C5a, and Ba); however, there appeared to be no clear clinical correlation between higher levels, and the authors suggest this might be due to a small sample size [46]. Another study by Aguirre et al. in Argentina evaluated both AChR antibody titers and C5a and C3a; although there was a correlation between C5a levels and increased myasthenia gravis composite (MGC) scores, there was no significant difference between controls and MG patients [45]. A more recent study by Huang et al. in 23 patients found increased levels of C5, C5b-9, and C3a, with C3a levels having the highest sensitivity and specificity [47]. It is possible that some of the differences in these studies derive from interpersonal differences between patients.

In a study evaluating the proteomic signature of 140 patients, Nekle et al. found four distinct proteomic profiles [48]. Protein signature 3 was associated with higher concentrations of C9 as well as increased MGC scores and was found to identify patients who were more likely to respond to complement inhibition therapies. As C9 is part of the membrane attack complex, it is plausible that the further validation of complement levels can result in the better identification of patients who might respond to complement inhibition therapies. There is further evidence that there might also be differences between thymomatous and non-thymomatous MG, which might affect responses to complement inhibition therapy [49,50]. Thymomatous MG has been associated with lower total C3 levels, indicating higher complement activation, which might have therapeutic implications [51,52]. It has been proposed that this is due to a higher prevalence of complement-activating autoantibodies against titin and ryanodine in thymomatous MG [49,53].

## 5. Clinical Use of Complement Inhibitors

### 5.1. Currently Available Therapies

Eculizumab is the first commercially available humanized monoclonal antibody complement inhibitor, first approved in 2007 for paroxysmal nocturnal hemoglobinuria and approved for generalized myasthenia gravis with AChR antibodies in 2017 [54]. Eculizumab acts on C5, preventing its cleavage into C5a and C5b [55] and inhibiting the activity of the membrane attack complex while preserving proximal complement products important for chemotaxis [56]. The binding of eculizumab (and both ravulizumab and zilucoplan) depends on the presence of an arginine residue in position 885; some patients may exhibit a mutation substituting this residue for a histidine (p.Arg885His), which grants resistance to the activity of eculizumab. Approval for eculizumab was based on results of the REGAIN Trial (Phase III randomized, double-blind, placebo-controlled trial), which randomized 125 patients with generalized non-thymomatous AChR antibody-positive MG who had failed two prior immunosuppressive therapies to either eculizumab or placebo [57]. The study showed no statistical significance on its primary endpoint; however, there was evidence of benefits in post hoc and secondary analysis, with benefits sustained up to 130 weeks [58]. Eculizumab has also been shown to improve Myasthenia Gravis Activities of Daily Living (MG-ADL) scores in patients with thymomatous MG [59].

Eculizumab is administered intravenously every 2 weeks after a loading phase of the doses in a month [60]. At 26 weeks, with every 2-week dosage, the mean plasma peak concentration was around 783 µg/mL, and troughs were around 341 µg/mL, well above the recommended minimum target of 50 µg/mL [60]. Steady state concentrations are usually achieved by 4 weeks. One study showed that after a first dose of eculizumab, free C5 concentrations at the trough level of eculizumab were <0.5 µg/mL in 57 of 62 studied patients. Given that plasma can provide a bolus of complement proteins, it is important that patients receiving eculizumab who undergo plasma exchange receive a supplemental dose of eculizumab [61]. This dose should be given within 60 min after plasma exchange (or within 60 min prior to fresh frozen plasma administration). Eculizumab can lead to improvement in MG-ADL scores as early as 1 month, although a period of 12 weeks is generally needed before considering a treatment failure [62]. With eculizumab as well as all complement inhibitors, patients should be vaccinated against Neisseria meningitides.

The second complement inhibitor approved for myasthenia gravis was ravulizumab, first approved for paroxysmal nocturnal hemoglobinuria in 2018 [63]. Ravulizumab acts on a similar mechanism as eculizumab; however, it was designed to have a longer half-life via four aminoacid substitutions: two substitutions largely eliminate the increased clearance of ravulizumab after binding to C5 and the other two increase the affinity for human FcRn without blocking it, leading to the recycling of the complex and increases in the ravulizumab half-life [64,65]. Target serum concentrations are achieved after one dose (>175 µg/mL) and sustained over a 26-week period, with terminal complement inhibition achieved after the first dose [66]. Differences in weight between patients do not seem to affect maximum and trough concentrations significantly.

Ravulizumab was approved after the CHAMPION-MG trial, a Phase III randomized, double-blind, placebo-controlled trial. Like REGAIN, patients with PLEX within the last 4 weeks and patients with thymoma or thymectomy in the past 12 months were excluded [67]. Both the MG-ADL (primary endpoint) score and Quantitative Myasthenia Gravis (QMG) score (secondary endpoint) at 26 weeks were significantly improved in the ravulizumab group when compared to placebo. On an open-label extension, benefits were sustained at 60 weeks [68], and a post hoc analysis showed similar responses in patients regardless of the time of diagnosis of myasthenia gravis [69]. Evidence for thymomatous MG patients appears to be more limited, with case report evidence showing patients may have a sustained response with ravulizumab after being treated with eculizumab [70], although to our knowledge, no other studies have evaluated the response of thymomatous MG patients to ravulizumab.

Zilucoplan, approved for AChR antibody + generalized myasthenia gravis in 2023 by the Food and Drug Administration (FDA), is another C5 inhibitor, peculiar in the fact that it does not represent a monoclonal antibody but a macrocyclic peptide [71]. It is administered once daily, with peak drug levels observed after 3 h, reaching steady state concentration in approximately 11 days. Body fat percentage and weight do not appear to affect pharmacokinetics significantly; however, the molecule and its metabolites are 99% bound to plasma proteins. By 1 week, complement inhibition reaches 97.5% [72].

The efficacy of zilucoplan was evaluated in the RAISE clinical trial, a Phase III randomized, double-blind, placebo-controlled trial. Similar to its predecessors, zilucoplan showed statistically significant improvements in MG-ADL scores (in this case, from baseline to week 12). The most common adverse event in the original trial was injection site bruising [72]. An open-label extension (RAISE-XT) evaluated treatment-emergent adverse events as a primary endpoint and MG-ADL as a secondary endpoint at 12, 24, and 60 weeks. The most common adverse events were MG transient clinical worsening, diarrhea, and COVID-19, with only one patient experiencing injection site infection in the right inner thigh (which was clarified by trial investigators as not being a recommended injection site) [73]. Of note, another analysis of the RAISE and RAISE-XT trials by Weiss et al. revealed that patients on zilucoplan were more likely to present improved fatigue based on the Quality of Life in Neurological Disorders Short Form Fatigue Scale [74]. To our knowledge, there have been no studies evaluating the use of zilucoplan in patients with thymomatous myasthenia or seronegative myasthenia.

Some studies have indirectly compared these three therapies in the context of myasthenia gravis. A systematic review and meta-analysis of 872 patients over 13 trials by Ma et al. compared different myasthenia gravis medications with the QMG change as the primary outcome. Eculizumab and zilucoplan ranked second and third, respectively, although there was no statistically significant difference between any of the medications studied (except for batoclimab, which ranked #1) [75]. Gordon Smith et al. compared the complement inhibitors with the neonatal Fc receptor blockers rozanolixizumab and efgartigimod, without significant differences in the improvement of QMG [76]. Overall, the preponderance of evidence suggests that there is no significant difference between the different complement inhibitors, and further research is needed to evaluate whether there is a significant difference between complement inhibitors and neonatal Fc receptor blockers [77].

### 5.2. Emerging Therapies

Other C5 inhibitors with different chemical structures are currently being studied for the management of myasthenia gravis. One of these is cemdisiran, a small interfering RNA that suppresses the hepatic production of C5. Cemdisiran is composed of a 21-nucleotide sense strand and a 25-nucleotide antisense strand conjugated to N acetylgalactosamine, which targets delivery to the liver via asialoglycoprotein receptors. Cemdisiran binds away from a common C5 mutation that grants resistance to eculizumab (p.Arg885His); thus, cemdisiran might be an option for patients with that mutation and resistance to eculizumab [78]. Studies in passive rat MG models show reduced weakness and a 10-fold reduction in C5 mRNA expression, a reduction in C5 levels to <20% with a single dose, and a reduction in hemolysis of 70% with a 5 mg/kg dose [79]. Similar positive results were seen with EAMGs. Further studies in non-human primates showed that the addition of another C5 inhibitor, pozelimab (which is able to bind to both wild-type and variant p.Arg885His), resulted in the better suppression of hemolysis in a paroxysmal nocturnal hemoglobinuria model, with neither agent alone achieving reductions in hemolysis of >80%, while the combination of both agents resulted in >80% inhibition sustained for 8 weeks with doses of cemdisiran and pozelimab of 5 + 5 mg/kg and 5 + 10 mg/kg and for 13 weeks with doses of 25 mg + 10 mg/kg [80]. Currently, the NIMBLE trial (Phase III randomized, placebo-controlled trial) is evaluating pozelimab + cemdisiran in both AChR and LRP4 myasthenia gravis patients, with the primary outcome being the MG-ADL score at 24 weeks. Like CHAMPION-MG and REGAIN, patients with MuSK antibodies or thymoma were excluded from the study.

Another C5 inhibitor currently being studied is gefurulimab. Gefurulimab differs from other C5 inhibitors in that it is designed to be given by subcutaneous injection, like zilucoplan; however, unlike zilucoplan, this medication appears to block C5 in both wild-type and mutant p.Arg885His patients [81]. The medication is currently being studied as a once-daily injection for AChR myasthenia gravis in the PREVAIL trial.

Besides C5 inhibition, other potential targets in the complement cascade have been investigated. C6 and C7 inhibition have been studied, with each of them posing theoretical advantages over C5 inhibition [82,83]. It has been suggested that some patients who are non-responders to eculizumab might have incomplete complement inhibition, which might be related to increased C3b density as well as to the p.Arg885His mutation. It is possible that many patients on eculizumab have some degree of residual complement activity; however, the exact clinical implication of this finding is uncertain. It has been suggested that C6 and C7 inhibition might be an acceptable alternative or complement to C5 inhibition [84].

C6 is present in blood at a concentration of 45–60 µg/mL, compared to 75 µg/mL in the case of C5. Unlike C5 inhibition, blocking C6 leaves C5a formation intact. Whether sparing C5a results in better or worse clinical outcomes in myasthenia gravis is unknown. A C6 complement inhibitor, mAb 1C9, demonstrated dose-dependent hemolysis in an experimental model, including CD59- red blood cells [85]. Interestingly, many C6 inhibitors have been developed experimentally, with most of them not exhibiting a significant effect on complement activation, suggesting that most epitopes in C6 do not result in the inhibition of the protein [86].

C7 antibodies offer similar theoretical advantages to C6 inhibitors. TPP1820, a mAb targeting C7, was protective in passive experimental myasthenia gravis rat models. The further evaluation of patient plasma in vitro recognized four subgroups with different responses to C7 inhibition, suggesting that different patients may have different responses to C7 inhibition, akin to observations with eculizumab [84].

Other molecules that have been studied include Factor B (targeting the alternative pathway of complement) [87] as well as Factor D (targeting the alternative pathway more proximally) [88]. Theoretically, the use of these medications would allow for inhibition even more upstream, which would lead to more profound complement inhibition; however, as mentioned earlier, this would also result in the blocking of both C3a and C5a, which have other immunoregulatory functions. Whether these approaches would result in improved clinical outcomes is uncertain.

As of the date of this writing, only vemircopan, a Factor D inhibitor, is being studied in clinical trials for myasthenia gravis (Phase II). Danicopan, a similar factor D medication, has previously been studied as an add-on therapy to eculizumab in patients with PNH74, with preliminary results showing improvement of hemoglobin concentrations with a favorable safety profile [89]. Table 1 summarizes the various complement-targeting therapeutic agents discussed in this article.

## 6. Conclusions

The current review summarizes the current evidence regarding the pathogenic role that complement plays in AChR antibody-positive myasthenia gravis. There is a need for further study of potential biomarkers for the evaluation of both treatment response as well as prognosis. Currently, complement-targeting effective therapies are available for myasthenia gravis, albeit with some limitations, including the possibility of individual resistance due to C5 mutations. Current research into other alternative therapies is likely to yield next-generation complement inhibitors that are likely to become a staple of future myasthenia gravis management.

## Figures and Tables

**Figure 1 cells-14-00739-f001:**
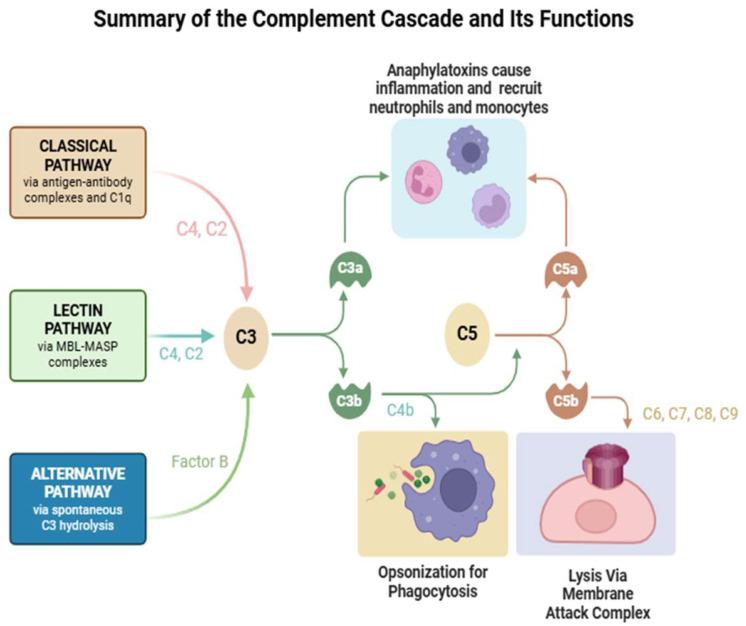
Summary of the complement cascade and its functions. Created in Biorender. Jaffer, M. (2025) https://BioRender.com/illustrations/67b87bf898d27ce5f197797a?slideId=37f77c61-0bc1-4db8-b97b-6e32c4ddd66c.

**Figure 2 cells-14-00739-f002:**
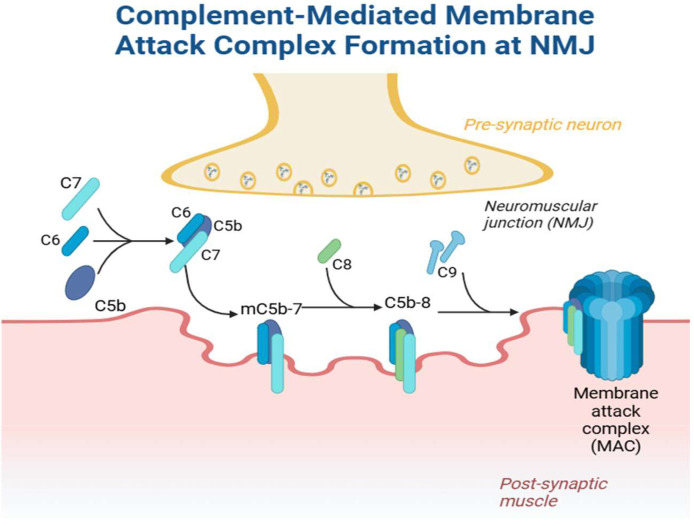
Complement-mediated membrane attack complex formation at the neuromuscular junction. Created in Biorender. Jaffer, M. (2025) https://BioRender.com/illustrations/67b8ca00cd23a08cdcf9decf?slideId=01ed1598-d900-4eff-aba0-8e4687b49cfd.

**Table 1 cells-14-00739-t001:** A summary of the various complement-targeting MG medications, mechanisms of action, target pathways, clinical trials, and evidence for use.

Medication	Mechanism of Action	Targeted pathway	Trials	Evidence
Eculizumab	C5 inhibitor, requires arginine in position 855	Common pathway	REGAIN (Phase 3)	AChR+ with generalized non thymoma associated MG, improve MG ADL scores at 12 weeks, sustained up to 130 weeks
Ravulizumab	C5 inhibitor, requires arginine in position 855	Common pathway	CHAMPION-MG (Phase 3)	AChR+ with generalized non thymoma associated MG, improved MG ADL scores at 12 weeks, sustained up to 60 weeks
Zilucoplan	C5 inhibitor, requires arginine in position 855	Common pathway	RAISE III (Phase 3)	AChR+ with generalized non thymoma associated MG, improved MG ADL scores at 12 weeks, sustained up to 60 weeks
Cemdisiran- pzelimab	siRNA targeting synthesis of C5, C5 inhibitor targeting both wild type and mutant R855H	Common pathway	NIMBLE (Phase 3, in progress)	AChR+ and LRP subgroups being studied
Gefurulimab	C5 inhibitor targeting both wild type and mutatnt R855H	Common pathway	PREVAIL (Phase 3, in progress)	AChR+ with generalized MG subgroup being studied
Vemircopan	Factor D inhibitor	Alternative pathway	NCT05218096 (phase 2 trial in progress)	AChR+ with generalized MG subgroup being studied
C6 inhibitors	Inhibition of C6	Common pathway	Not in trials	N/A
C7 inhibitors	Inhibition of C7	Common pathway	Not in trials	N/A
Factor B Inhibitors (IONIS-FB-L-Rx)	Factor B inhibition, antisense oligonucleotide	Alternative pathway	Phase 3 trials for IgA Nephropathy in progress	N/A

## Data Availability

No new data were created or analyzed in this study.

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
