# Peer review of "The Role of Complement in the Pathogenesis and Treatment of Myasthenia Gravis"

_cells, 2025, doi:10.3390/cells14100739_

Round 1

Reviewer 1 Report

Comments and Suggestions for Authors

This is a comprehensive review on the role of complement in the pathogenesis of myasthenia gravis and current treatments targeting complement and they end the review with some emerging therapies.

The paper is well-written and the bibliography is updated.

Comments:

Regarding the mechanisms of autoantibodies involving complement activation I would suggest to include  a paper of 2022 (PMID: 36074148) in which the authors demonstrate that the combination of antibodies targeting disparate subunits of the acetylcholine receptor elicit a stronger complement activation. They suggest that this synergistic interaction can explain the well-known discrepancy between serum anti-AChR titers and clinical severity.

Reference 76 is wrongly cited. The authors refer to rozanolixizumab and efgartigimod as complement inhibitors but they are blockers of neonatal Fc receptors.

Author Response

Comments 1: "Regarding the mechanisms of autoantibodies involving complement activation I would suggest to include  a paper of 2022 (PMID: 36074148) in which the authors demonstrate that the combination of antibodies targeting disparate subunits of the acetylcholine receptor elicit a stronger complement activation. They suggest that this synergistic interaction can explain the well-known discrepancy between serum anti-AChR titers and clinical severity."

Response 1: Thank you for your review! Regarding the first comment, we have included the reference as reference 40.

Comment 2: "Reference 76 is wrongly cited. The authors refer to rozanolixizumab and efgartigimod as complement inhibitors but they are blockers of neonatal Fc receptors."

Response 2: Thank you for noting the imprecise wording. We have changed the language to make it more clear. Changes are highlighted in red on the revised manuscript.

Reviewer 2 Report

Comments and Suggestions for Authors

This is very well done review of complement mechansims is myasthenia gravis.  My only editorial suggestion is that section 3 ( other complete mediated disorders) seems outside of the scope of the paper and should be deleted.

Author Response

Comment 1: "My only editorial suggestion is that section 3 (other complete mediated disorders) seems outside of the scope of the paper and should be deleted."

Response 1: Thank you very much for your review! We took your comment under consideration and after discussing among our co-authors, we would like to opt to maintain this section in the manuscript, because we believe it is important demonstrate and discuss other pathogenic processes mediated by complement disorders. This helps to better contextualize MG in the spectrum of autoimmune diseases.